# Factors influencing decisions about whether to participate in health research by people of diverse ethnic and cultural backgrounds: a realist review

Eleanor Jayne Hoverd [1], George Hawker-Bond,[2] Sophie Staniszewska,[3] Jeremy Dale [3]

¹University of Warwick, Coventry, UK
²Nuffield Department of Clinical Neurosciences, Oxford University, Oxford, UK
³Warwick Medical School, University of Warwick, Coventry, UK

**Correspondence to**
Eleanor Jayne Hoverd;
eleanor.hoverd@nihr.ac.uk

## ABSTRACT

**Objective** To develop and refine a programme theory that explains factors that influence decisions to take part in health research by people of diverse ethnic and cultural backgrounds.

**Design** Realist review following a sequence of five steps: (a) scoping search and identification of programme theory; (b) evidence searching; (c) critical appraisal and data extraction; (d) organisation of evidence and (e) refinement of programme theory.

**Eligibility criteria** Documents (including peer-reviewed articles, grey literature, websites, reports and conference papers) either full text, or a section of relevance to the overarching research question were included.

**Data sources** EMBASE, Medline, Web of Science, Psych Info, Google and Google Scholar were searched iteratively between May and August 2020. Search strategy was refined for each database providing a broad enough review for building of programme theory.

**Analysis** Data from eligible documents was extracted to build understanding of the factors that influence decision-making. Data were mapped to create a data matrix according to context (C), mechanism (M), outcome (O), configurations (C) (CMOCs) for the process of informed consent, to aid interpretation and produce final programme theory.

**Results** 566 documents were screened and 71 included. Final programme theory was underpinned by CMOCs on processes influencing decisions to take part in research. Key findings indicate the type of infrastructure required, for example, resources, services and policies, to support inclusion in health research, with a greater need to increase the social presence of researchers within communities, improve cultural competency of individuals and organisations, reduce the complexity of participant information, and provide additional resources to support adaptive processes and shared decision making.

**Conclusion** The review indicates the need for a more inclusive research infrastructure that facilitates diverse participation in health research through incorporating adaptive processes that support shared decision making within the informed consent process and in the conduct of research projects.

## STRENGTHS AND LIMITATIONS OF THIS STUDY

⇒ Using realist methods to explore the contexts and mechanisms of the complexities affecting the informed consent process in health research, for people of diverse ethnic and cultural populations, allowed new insight to emerge.

⇒ There is a paucity of evidence around the informed consent process in health research with people of diverse ethnic and cultural backgrounds, limiting the evidence from which to extract data from.

⇒ Relevance and rigour were increased through the involvement of National Institute for Health Research (Public) Research Champions, healthcare professionals and clinical academics in refining the programme theory.

⇒ Owing to the COVID-19 pandemic, stakeholder involvement in the programme was conducted remotely, enabling access from a wider geographical spread of stakeholders.

⇒ Whilst the review was undertaken in the context of the UK health care system, the programme theory is likely to have wider applicability.

## INTRODUCTION

People who have poorer health outcomes, and have greater needs are currently underserved by the way in which health research is accessed and delivered.[1] Lack of diversity of participants taking part in health research, poses a serious challenge both within the UK and globally, around how to address and develop a more inclusive health research system so that under-served populations can benefit. It has been argued that the informed consent process may be contributing to inequalities in health research through a lack of flexibility in the process.[2–10] The informed consent process in health research aims to provide patients and the public with the information they require to make a voluntary decision about participation in health research. In May 2019, the International

Council for Harmonisation[11] drafted the *General Considerations for Clinical Studies E8 (R1)* suggesting that a 'one size fits all approach' to studies should be avoided. Hence, it is important to unravel the factors that may affect the decision to take part in health research by these populations, to develop a system that is more inclusive.

Evidence suggests that the informed consent process, including the language used within it, creates a serious barrier to access and participation.[8 12–17] Several systematic reviews highlight common barriers to the informed consent process and indicate the need for further research to determine how information is delivered and what information people need, to make an informed decision about participation in health research.[18–20] There is less focus in the literature around what works, to inform research design and delivery. It is a critical issue that requires considerable improvement to develop a more inclusive health research system.

Conducting a realist review of the informed consent process in health research with under-served populations, is a well-suited method for exploring such a complex intervention[21] to explain how the informed consent process may work for whom, how and under what circumstances. The overarching research question for this review is: What contextual factors influence the decision to take part in health research in under-served populations?

The term under-served is preferred when referring to groups or communities who are under-represented in health research, for example, marginalised groups, and is also used to describe geographical locations such as rural or low-income countries, although a single definition is not found within the literature.[1] There are a number of intersecting factors that contribute to groups being under-served; such as demographic, socioeconomic, health status and disease-specific status, leading to disadvantage and discrimination.[1] However, being under-served is also likely to be context-specific, with key characteristics related to trial design influencing the ability to participate.[1] For example, the informed consent process is recognised as a specific barrier to inclusion in health research by under-served populations.[1]

Objectives were as follows:
1. To undertake a realist review to identify the contexts and mechanisms that affect the informed consent process and the decision to participate in health research in under-served populations.
2. To draw on the review's programme theory to develop guidance that addresses barriers to participating in health research in under-served populations.

## METHODS
In this study, we undertook a realist review to produce a programme theory to explain the informed consent process, and hence provide actionable recommendations for policymakers and health researchers to inform the development of inclusive health research. The review process followed the steps as laid out in the protocol[22]

and was conducted and reported in accordance with RAMESES (Realist and Meta-narrative Evidence Syntheses: Evolving Standards) guidance and publication standards, as well as providing a rationale for selecting this approach.[23] RAMESES guidance has been produced to provide initial reporting standards for realist reviews and evaluations.[23]

Following a realist analytical process, context, mechanism, outcome, configurations (CMOCs) were built by EJH. These were shared and discussed at meetings with the review team and stakeholders whereby feedback was encouraged which contributed to further refinement and establishing a long list of CMOCs. The CMOCs conceptualise key features that support explanation and understanding of complex programmes and create a model that describes how mechanisms are activated, for who and under what circumstances, which affects outcomes.[23] Summaries were developed after each meeting to support reflection and understanding with stakeholders. Diagrams were constructed and refined to support explanation of the findings.

### Patient and public involvement
The National Institute for Health Research (NIHR) funds health and social care research in the UK and advocates for patient and public involvement (PPI) throughout all stages of research. To support PPI, the NIHR has introduced the role of NIHR (Public) Research Champions who are patients or members of the public, and volunteer to raise awareness of health and social care research as well as help researchers to understand the participant experience.[24] Six NIHR (Public) Research Champions were involved in this review, helping to clarify the scope of the review, contributing to the development of programme theory and developing recommendations. Three of the NIHR (Public) Research Champions were from populations of diverse ethnic and cultural backgrounds.[24] Their involvement is discussed in more detail (step 1).

### Development of an initial programme theory (step 1)
In order to develop an initial programme theory, a scoping search (see online supplemental file 1—Scoping searches) was carried out based on search terms centred on the intervention under study (informed consent) with under-served populations in health research, to explore key literature in this field. This allowed us to explore the available literature and to create some initial boundaries.[25] Theoretical concepts and evidence that would contribute to explaining different outcomes; for example, positive (understood study information) or negative experiences (excluded from taking part). A theoretical concept map was created (figure 1) identifying higher-level, abstract theories which provided a basis to discuss some of the larger complexities of informed consent with the stakeholder group in combination with the empirical evidence from the scoping search, as a way of supporting, contradicting or modifying the programme theory and to refine the purpose of the review.[25]

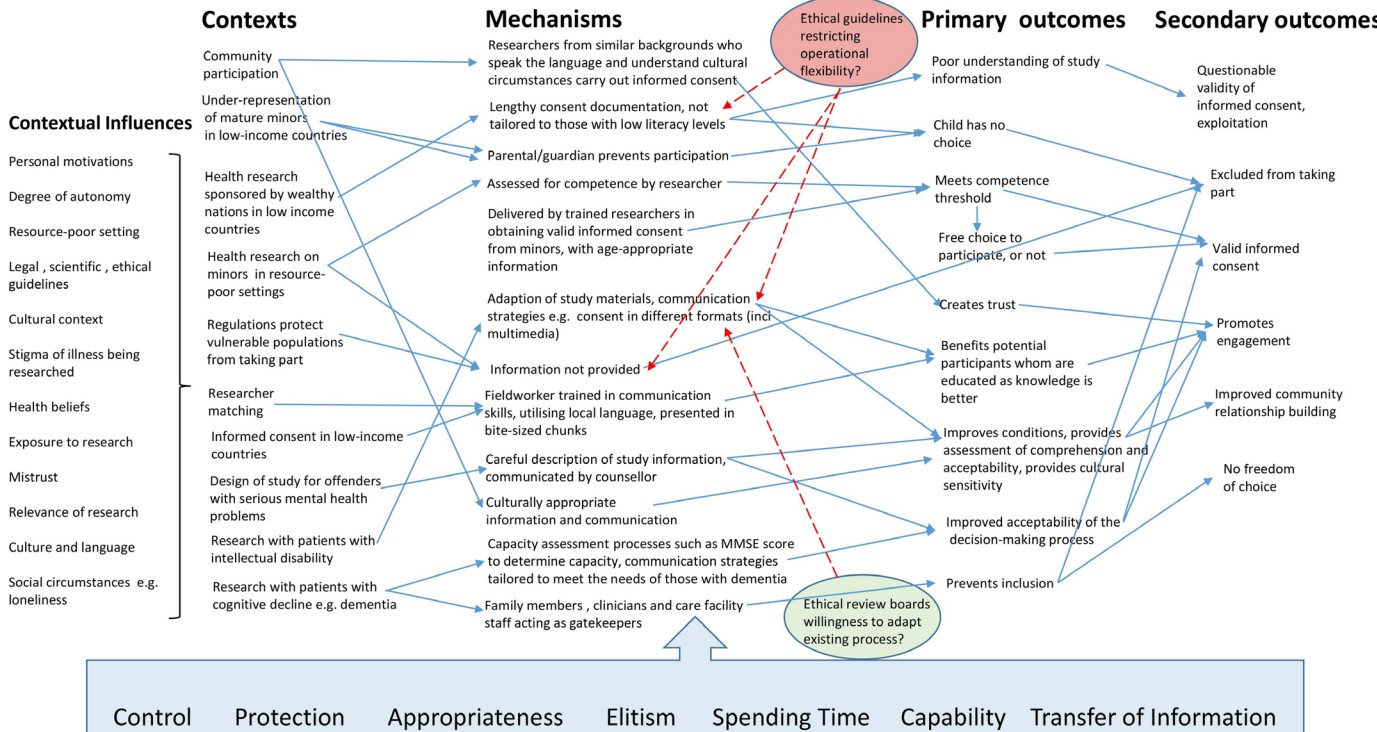

**Figure 1** Theory map—factors influencing decisions about whether to participate in health research with under-served populations—identified from scoping search.

The initial scoping search resulted in 16 documents selected based on relevance to the overarching research question.[4 26–37] These were examined, with codes applied to label CMOCs, concepts and theories, which were helpful in building the initial programme theory. These CMOCs were then collated, a long list of CMOCs drawn-up and then developed into a diagram, to create a visualisation of the initial programme theory (figure 2). This enabled identification of key theories for further exploration.

### Stakeholder involvement

Stakeholders included NIHR (Public) Research Champions, health professionals (research nurses, a dietician, physiotherapists and a general practitioner) and clinical

**Figure 2** Theoretical concepts compounding positively, or negatively, the intervention.

academics who were openly invited to participate via the review team's associated Medical School and NIHR Clinical Research Network. A total of six meetings were held virtually (three with NIHR (Public) Research Champions and three with the mixed group of health professionals and clinical academics). Stakeholders were geographically widespread and thus virtual meetings enabled easier access than had meetings been held face to face. At each stakeholder meeting, background information was provided, including an explanation of realist methodology. The facilitated discussions covered under-served populations; initial programme theory; analysis; results and recommendations; and stakeholder feedback and advice.

Meetings were organised to provide perspectives and expert opinions around issues relevant to building programme theory, insights into the ways in which social structure may influence the informed consent process, as well as identifying any pertinent documents of relevance.[21] Stakeholder input is described in the protocol.[22]

### Evidence searching (step 2)

Literature searches were conducted iteratively and conducted between May and August 2020 (see online supplemental file 2—Searches). The initial search used the databases EMBASE, Medline, Web of Science and Psych Info, in addition to manual-searching using Google and Google Scholar. The search strategy was refined for each database, aiming to provide a broad enough review for further building of programme theory.

Following the initial search, 468 citations were identified following removal of duplicates, with 54 citations remaining following first and second level screening by EJH and GH-B. As synthesis progressed, key theories were emerging around the informed consent process with people of diverse ethnic and cultural backgrounds, with most of literature from the USA. Therefore, following discussion with stakeholders and the review team, it was agreed synthesis would focus on a smaller number of key theories with additional searches required to test and refine programme theory, focussing on a 'conceptually rich'[38] sub-group of literature to produce recommendations relevant to the UK health research system. The reason for limiting the subsequent searches to test theories within the UK literature was because an inclusion policy is lacking, and with a free, publicly funded healthcare system, context and resources may differ in comparison to other geographical locations, thus affecting mechanisms and outcomes. A flow diagram of the iterative searching process is displayed in figure 3, based on RAMESES guidelines.[23]

A key consideration of a realist review is to reflect on contextual factors of the social system(s) and how this affects the working of the intervention.[21] In developing the initial programme theory, we sought to view the informed consent process through the larger social system at micro (individual), meso (organisational, institutional) and macro (policy) levels, where interactions between individuals and levels can be understood better.[39] This was aimed at bringing about recommendations for change.[20]

### Selection and appraisal of documents (step 3)

Broad inclusion and exclusion criteria (see online supplemental file 3—Inclusion/Exclusion criteria) were applied

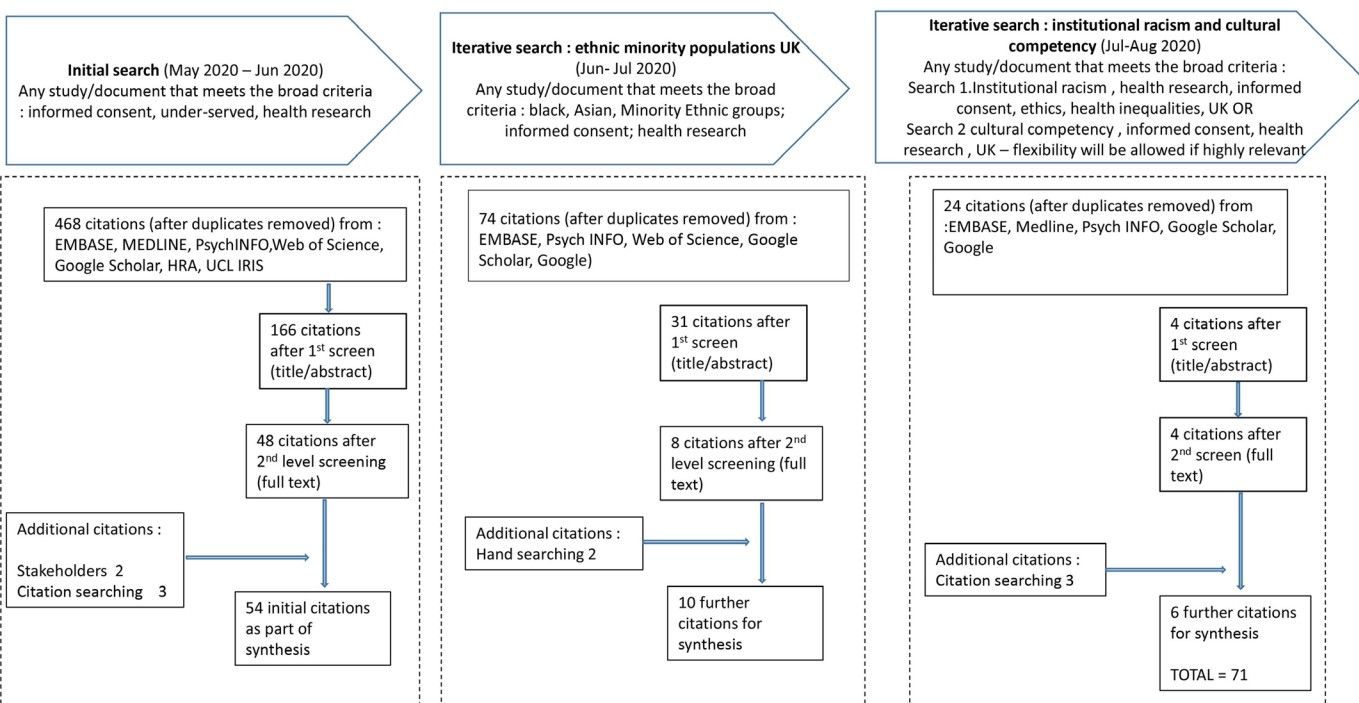

**Figure 3** Flow diagram of searching.

in the selection and appraisal process, each document was assessed for relevance and rigour by EJH and GH-B, independently.[40] Relevance and rigour were assessed in parallel. Decisions were based on whether a document was relevant to the overarching research question and what it contributed to the review. Rigour was determined by assessing methodological rigour and whether a document included important information that may help with interpretation, as well as providing sufficient detail to determine trustworthiness.[23] Quality of data was noted in data extraction forms and considered during analysis and synthesis. All relevant data were used to build the arguments that supported the final programme theory, including systematic and literature reviews.[25] Two papers included in the review are noted to be included in a systematic review, but do not impact on the review in terms of the importance given to CMOC development.[6 41] Documents identified as being helpful in developing, refuting and refining the programme theory were sought, although the number of iterations and length of time spent on this was limited due to the timescale of the review.[25]

EJH screened all documents identified through searches, by title and abstract. Second reviewer GH-B screened a random 10% of the documents.[42] To establish inter-reviewer agreement, a kappa measure of k>0.8 was predetermined, with a resultant kappa measure of 0.90 following second level screening by EJH and GH-B.

### Data analysis and synthesis (step 4)

A combination of annotations and notes, along with a data extraction form was used to classify information from documents about potential candidate theories, through noting CMOC concepts.[20] A data extraction form was adapted from Rycroft-Malone *et al*[43] to record characteristics of documents and whether evidence was good enough and relevant to include in the synthesis. This included author, year, source, issues of sample size, data collection and data analysis, and interpretations made, characteristics of theoretical concepts and the impact these characteristics may have on the informed consent process (eg, characteristics of appropriateness of informed consent and how these characteristics may impact on the decision to take part in research). The data extraction form was tested on the first three documents to confirm its applicability and usefulness.

### Refinement and validation of programme theory (step 5)

In step 5, both data and annotations were combined. Data synthesis was undertaken by EJH through mapping data to create a data matrix according to CMOCs which helped with interpretations and in developing theory.[23] Contexts were themed according to the level of the social system within which they related to.[39] Through noting common features, or relationships and identifying any underlying concepts, recurring themes were identified and patterns in the evidence were tested.

Recurring contexts and outcomes were explained through mechanisms. For example, within the included texts, the design of some research studies were reported as resulting in exclusion of people of diverse ethnic and cultural backgrounds, leading to mistrust, feelings of being exploited and a belief that academic communities are not committed to understanding the needs and experiences of people from diverse ethnic and cultural communities.[5 37 44–46] During data synthesis the aim was to provide an explanation of these contexts and outcomes through the identification of mechanisms. Further studies were sought to test the various elements of programme theory. Synthesis results were shared and discussed at stakeholders' meetings, to ensure validity and consistency of the interpretations made. The extraction of data, analysis and synthesis process was iterative with documents repeatedly read at various points throughout the process.

An overview of theories is described in the Results section, through a narrative of how the informed consent process works, for whom, how and under what circumstances based on the evidence in this review. References are made to CMOCs, including some quotations to draw out key aspects of CMOCs.

### Development of recommendations (step 6)

Recommendations were developed with stakeholders through presenting the findings and a narrative of the programme theory through a virtual meeting. EJH led the remote discussion leading to agreement on a set of practical recommendations aimed at government agencies, Research Ethics Committees, researchers and health professionals. Input from stakeholders was intended to ensure their practicality.

## RESULTS
### Document characteristics
In total, 71 documents published between 2005 and 2020 were included from 19 countries with the majority from the USA (37%), UK (24%) and Africa (13%). The date range of the sources was from 2004 to 2020. The source type was mixed including discussion papers and opinion pieces (17, 24%), qualitative research (27, 38%), literature reviews (4, 6%), mixed methods research (2, 3%), systematic reviews (5, 8%), quantitative research (9, 13%), guidelines (2, 3%), case study (1, 1%), book chapter (1, 1%), audit (1, 1%), report (1, 1%) and a working paper (1, 1%).

### Context, mechanism, outcome, configurations
The key themes that emerged related to inter-relationships; organisational and individual cultural competency and adaptive processes. Patterns in mechanisms were identified as can be seen on the long list of 29 CMOCs used to develop programme theory (see online supplemental file 4—CMOC long list). These are discussed below making reference to the associated, numbered CMOCs

and selected quotations from documents which help to explain the theories.

## Inter-relationships

Developing trusting relationships between researchers and under-served communities is important. Key family members, community or faith leaders and local health-care staff may influence a potential participant's decision to take part (CMOC 3, CMOC 5, CMOC 11).[3 47 48] The approach to identifying key individuals must be done sensitively with clear information provided about what participation involves (CMOC 1, CMOC 7, CMOC 9, CMOC 18, CMOC 26).[3 6 17–49] This can lead to increased autonomy, trust and a positive view towards health research with the key individuals becoming a trusted point of information, thus helping to improve knowledge about health research, increasing the awareness and confidence of communities to participate in research.[3]

> … research assistants attempted to immerse themselves in the activities of the diabetes clinic …[6] (P.7) (CMOC 1)

Researcher-matching may help (CMOC 1, CMOC 4) as it provides some choice for the potential participant as to whom they work with[7 50] and can enable more effective and positive experiences. This has been used in trials on diabetes, mental health and cardiovascular disease with participants of South Asian heritage and Black British, African and African-Caribbean heritage.[51 52] There does not appear to be one definition of researcher-matching; examples include: an interviewer speaking the same language as the participant conducts interviews; a health professional who speaks the same language as participant conducts intervention; an interpreter is present to support delivery of the intervention; trained multi-lingual facilitators; ethnically matched outreach team delivers the intervention; research assistants who speak the same language; and culturally adapted interventions.[2 9 17 37 47 52–58] The cost implications of this approach were rarely reported, with only one study discussing the impact and challenges of training facilitators and the complexities of translating material.[9] The perceptions of having interviews or focus groups conducted by someone of the same ethnicity is under-explored and viewed by some researchers as not as essential as language, gender, technical and interpersonal skills which may be more important.[9] There was some suggestion that researcher-matching should be used with caution as there is a risk that this may lead to researchers lacking cultural competence.[59] There is also a risk when using interpreters who have been trained to facilitate qualitative interviews that the richness of data could be compromised.[60]

Family relationships may trigger a positive or negative experience of the decision-making process (CMOC 5, CMOC 11).[2–4 61] Identification of key family/community members involved in decision-making process can determine whether someone takes part and conversely can improve the choice through improved understanding (CMOC 5).[2 3 59 61]

## Organisational and individual cultural competency

There is considerable evidence to support the importance of delivering services with cultural competence (CMOC 2, CMOC 9).[3 5] Cultural competence is defined as the way in which behaviours, attitudes and policies of a system, or among individuals, such as healthcare professionals, enables a service to operate effectively among diverse cultural situations.[62] Developing cultural competency and leadership helps to develop a capable, skilled research workforce (CMOC 1)[3 5 7 10] protecting the autonomy of potential participants, encouraging reflection on the individual's life situation and developing awareness of different cultures.[5]

Contextual factors such as the life situation of some populations (eg, migrant communities) can influence the decision-making process in health research due to

> … fear of identification and reporting to authorities.[5] (P.13) (CMOC 11)

While cultural competence is evidenced to improve the responsiveness and appropriateness in the way in which health research is delivered, as well as reducing health disparities, it requires further exploration that goes beyond the scope of this review.[3 5 7 10 17] There is some evidence to suggest that cultural competency conflates language, geographical origin, ethnicity and race, potentially diluting the identity of individuals and communities.[63] The term has caused considerable debate since the introduction of the concept in the 1980s[62] with some asserting that it is not possible to be competent in other individuals' cultures.[58 64] However, cultural competence has been considered vital for engaging with diverse ethnic and cultural communities to ensure accurate information is shared appropriately to instil confidence in the safety and efficacy of COVID-19 vaccines.[65] Cultural competency when positioned within a human rights framework is viewed positively as a way of operationalising respect, providing a service that is culturally appropriate.[63 66]

## Adaptive processes

Providing small bitesize participant information about an available health research study, in local languages, with local cultural context supports understanding.[5 17 67] In addition to this, coproduction and more community-based participatory approaches may encourage wider participation, offering a way to empower individuals and communities who are under-served in health research.[68] It has been demonstrated that engagement with people of African-Caribbean heritage in the development of a culturally adapted family intervention for people with schizophrenia can be achieved by taking a two-way approach, whereby researchers become integrated within communities and individuals from communities are able to access training and support around health research to increase capability as part of a community

based participatory research approach, this has been demonstrated.[69]

Taking these approaches improves accountability and makes provided information more relevant.

… members of target populations should be involved in the overall design of research ….[41] (P.5) (CMOC 9, CMOC 14)

In addition, it may be helpful to offer a range of materials (eg, written, oral, audio and multi-media) to ensure concepts are understood (CMOC 2, CMOC 12).[3 28] Providing study information in various formats can promote feelings of independence, leading to consent being given freely, serving more respect for cultural differences (CMOC 2, CMOC 14).[6 7 70]

… this way you won't need to depend on your children anymore.[6] (P6) (CMOC 3)

Some groups lack an agreed written form of their main language, and alternative formats are useful in this respect (CMOC 2, CMOC 12).[6 7]

This creates trust and ensures dignity.[3 5 6 10 17 51]

### Institutional barriers

When the needs (eg, cultural, linguistic or historical) of people from diverse ethnic and cultural backgrounds are misrepresented or ignored[3 5 71] (CMOC 6, CMOC 8, CMOC 13) this results in: discriminatory outcomes (eg, exclusion due to eligibility criteria) and enforces inappropriate ethical, legislative, scientific guidelines and recruitment processes (CMOC 5, CMOC 13).[3 5 8] It has been recognised that individuals from diverse ethnic and cultural backgrounds are under-represented in commissioning and making funding decisions about health research, and on editorial boards of journals (CMOC 13).[9]

Study designs can restrict access to health research through creating eligibility criteria that excludes populations from taking part, usually due to a language barrier.[3]

… many studies require participants to have the ability to understand and speak English which excludes older people from ethnic minority communities and/or migrants.[3] (P.6) (CMOC 5, CMOC 9, CMOC 12, CMOC 13)[3 71 72]

This results in under-representation and may be perceived as institutional racism (CMOC 5, CMOC 9, CMOC 12, CMOC 13).[2 8 57 72]

… dearth of research with ethnic minority communities in the UK is a reflection of the lack of influential policy ….[3] (P.9) (CMOC 6, CMOC 13).

Evidence from the USA suggests that institutional racism is not easily defined and encompasses not only individual prejudices, but prejudices deeply embedded within the law, practice, economy, culture and society which result in unequal access to healthcare, including health research.[73] The role of institutional racism in health research may be fuelled by the costs for translation and additional culturally sensitive resources affecting the chances of receiving funding for grant applications.[52] However, there is little evidence available on the extent to which institutional racism occurs in health research. A systematic review found that it is a concept that is rarely explicitly named in abstracts and titles and rarely engaged with; lack of acknowledgement of its existence could be holding back discovery within this field.[63–74]

### Programme theory

A resultant programme theory was produced following the five steps shown in figure 4. The programme theory informed discussions with stakeholders to develop recommendations and actions (table 1) for individuals, organisations, institutions and policymakers to improve the informed consent process.

## DISCUSSION
### Statement of principal findings

This realist review of 71 sources summarises and expands on the existing evidence around decision-making and participation in health research through focussing on the context and mechanisms surrounding decision-making and understanding. The evidence base on participation in health research has tended to focus on the psychosocial drivers of participation, but use of a realist approach enabled greater consideration of contextual factors that influence whether participation occurs.

Through deeper exploration using iterative literature searching, data emerged that indicated the use of complex study documentation for example, patient information leaflets, communication barriers, lack of social presence in the community by researchers and a lack of diversity among researchers as key realist mechanisms which led to an explanation of the interaction with context to influence outcomes.[2 8 54 71] While the review team itself was not ethnically diverse, the stakeholder group were ethnically, professionally and organisationally diverse. EJH is part of an Equality, Diversity, Inclusion working group which supported engagement with patient and public stakeholders from diverse cultural and ethnic backgrounds.

At the individual level (micro) level, empowerment through access to resources, such as culturally adapted information, and the opportunity to be involved in the design of health research form a significant aspect of the programme theory. At the interpersonal and institutional (meso) levels, enabling shared decision making can result in positive outcomes through approaches such as coproduction; building community relationships and provision of researcher-matching and/or culturally competent researchers provides assurance and increases trust.

The differences and similarities between the use of the term shared decision making in clinical care and within a health research context would benefit from additional exploration of the literature. The term shared decision making is defined by the UK's National Institute for

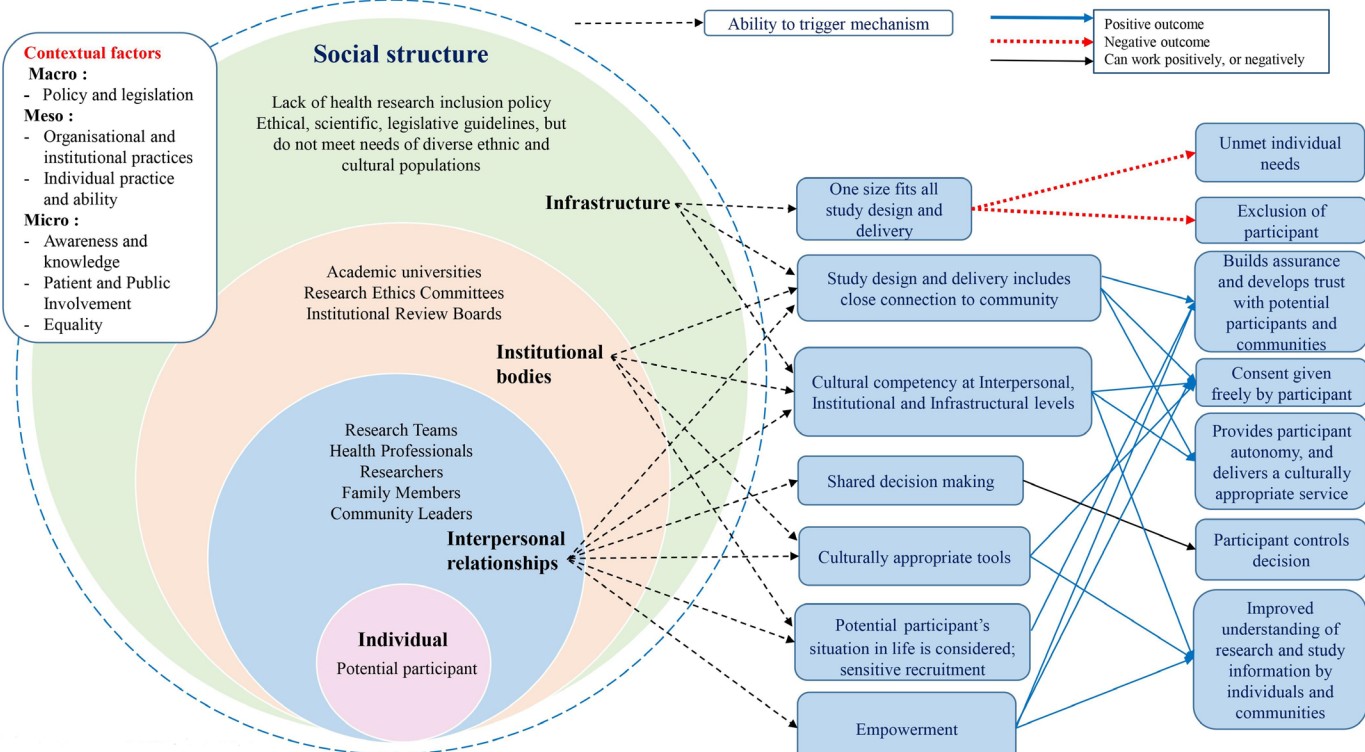

**Figure 4** Final programme theory.

Health and Care Excellence shared decision making collaborative as:

a process in which clinicians and patients work together to select tests, treatments, management or support packages, based on clinical evidence and the patient's informed preferences. It involves the provision of evidence-based information about options, outcomes and uncertainties, together with decision support counselling and a system for recording and implementing patients' informed preferences.[75 76]

It is of key importance in protecting patient/participant autonomy and voluntariness.[77 78]

In the context of this review, shared decision making is part of the informed consent process and requires establishing understanding by the potential participant, discussion of the risks and benefits, as well as clarifying

**Table 1** Recommendations

| Recommendation | Action required |
|---|---|
| 1. Policymakers, organisations, institutions and individuals need to take responsibility and address institutional racism, identifying and tackling issues that contribute to this (eg, lack of culturally adapted information, practices that may be contributing to institutional racism) | This requires:<br>► Leadership<br>► Identification of best practice<br>► Open discussion of culturally sensitive research ethics throughout the social system (eg, values, beliefs and practices)<br>Organisations and institutions should inform and educate their staff about institutional racism, what it is and how to address it. |
| 2. Build relationships with communities from diverse cultural and ethnic backgrounds to provide opportunities for patient and public involvement (PPI) and empower individuals so that study designs become more relevant and culturally appropriate | Employ more staff from diverse ethnic and cultural backgrounds in academic institutions and health research organisations.<br>Ethics committees should seek community members from diverse ethnic and cultural backgrounds.<br>Provide cultural competency training for all involved in health research.<br>Identify local champions and/or leads to raise awareness of cultural competency in health research. |
| 3. Develop tailored informed consent in health research that offers choices (eg, written, audio, multi-media), which may require additional resources | Ensure greater patient and public involvement (PPI) by people of diverse ethnic and cultural backgrounds in study designs. |
| 4. Organisations and institutions need to develop cultural competence within them, increasing capability, knowledge and awareness of other cultures | This requires cultural competence training that covers capability, knowledge and awareness in relation to research participation.<br>All who are involved in health research should undertake cultural competency training. |

the consequences of different options available.[78] Shared decision making is also a vital element of coproduction, or the involvement of patients and the public in shaping research projects.[79] In some instances, shared decision making can result in negative, or positive outcomes; for example, the hierarchies within some families may control decision-making through preventing or granting permission for family members to take part in health research.[6 9] More positive, shared decision making occurs when researchers spend time with communities, providing opportunities for informal discussions about research, building trust while developing cultural sensitivity to the 'norms and values' of particular groups.[26]

Resources at the macro level are lacking, such as the lack of flexibility in approaches to the informed consent process, leading to negative outcomes with needs not being met and exclusion from health research, which may be perceived as institutional racism.[46 52 57 74]

The data synthesis together with the stakeholder group meetings framed an overall programme theory in realist terms as:

> *If* organisations, institutions and individuals work together with diverse ethnic and cultural communities and become culturally competent, provide adaptive processes and empower community members to become involved, *then* this will develop and result in trust, better engagement, shared decision making and improved understanding.

The programme theory from this realist review is likely to have wider applicability despite being carried out in the context of the UK healthcare system. The informed consent process is a universal cornerstone to ethical practice in health research although may vary in the way in which it is carried out. However, rigid study designs and delivery may be contributing to the global issue of under-representation of under-served populations in health research.[18 50] While the critical issues described in this review are likely to be suitable for consideration by health research systems outside of the UK, it is recognised that resources and context differ across research systems, which may limit the transferability, particularly in low-income countries. The recommendations in this realist review requires evaluation in different settings to understand their wider applicability and transferability.

### Strengths and weaknesses in relation to other studies

Using a realist approach enabled a wider overview of more hidden factors that affect the decision-making process in health research, regarding the informed consent process with people of diverse ethnic and cultural backgrounds. The realist review has drawn from both peer-reviewed literature (66, 93%) and grey literature (5, 7%) as well as bringing in expertise from the stakeholder group. The lack of literature available on health research in general with people of diverse ethnic and cultural backgrounds is a weakness as more specific evidence about what works, for whom and under what circumstances is urgently

needed in order to help researchers design studies that are more inclusive and inform service development.[3 5] It is acknowledged that not all sources may have been discovered throughout this realist review process, in addition to not including the expertise of policymakers due to restrictions on time.

The review was conducted in accordance with RAMESES quality and publication standards to ensure transparency and is viewed as a strength.[19 43]

The involvement of NIHR (Public) Research Champions, healthcare professionals, clinical academics and PPI lead in refining the programme theory was felt to enhance the review's relevance. Having a second reviewer can increase the number of relevant studies included in a review and therefore this was considered a strength.[5]

### Suggestions for further research

From the review, cultural competence emerged as a recurring mechanism within the literature with further gaps in the evidence around: the way in which studies are designed and delivered and how people from under-served populations are involved in this aspect of the research process; institutional racism and its impact on health research; experiences of participation in health research with people of diverse ethnic and cultural backgrounds; the best adaptive informed consent processes, particularly for people of diverse ethnic and cultural backgrounds. In addition to this, further research is required to understand more the full range of requirements that may be needed to develop a more inclusive informed consent process, alongside in-depth enquiry around cultural competence and researcher-matching in relation to outcomes about research participation.

### CONCLUSION

The resultant programme theory explains how, when, why and for whom the processes involved in consenting to participate in health research does and does not work, and leads to suggestions for improvements to achieve greater inclusivity with people of diverse ethnic and cultural backgrounds.

Some aspects of the programme theory (eg, institutional racism, cultural competency in health research, researcher-matching) need more research to better understand how they affect outcomes. A critical finding is that a research infrastructure that supports and mandates inclusion is needed, as well as additional resources to support adaptive processes, support shared decision making through involving more patients and the public in the design of health research studies. A greater focus on the evidence base developing in the field of global health community engagement, particularly around coproduction, may provide valuable insights into how patients and the public from under-served communities can become involved in the design of health research studies.[79]

**Acknowledgements** We thank Samantha Johnson Information Specialist, for critically appraising the search strategy. We would also like to thank the following stakeholders involved in this review for their contribution to improving the quality and relevance of the final programme theory as well as the recommendations: Anne Devrell, Geoff Robson, Ifeanyi Sargeant, Mohini Samini, Dr. Mohammed Shaikh, Richard Stephens, Sandra Prew, Dr. Farhana Lockhat, Dr. Emma Sutton, Zi Liew, Sarah Joshi, Claire Talbot, Jonathan Heffernan-Davies, Sylvia Turner, Mohamed Mooradun.

**Contributors** EJH was the lead reviewer. GH-B completed second reviewer tasks. SS, JD, GH-B contributed to interpretation, critically reviewed and edited the manuscript. EJH was responsible for the initial design and drafting of the manuscript. All authors read and approved the final manuscript. EJH acts as guarantor.

**Funding** This work was supported by a National Institute for Health Research (NIHR), HEE/NIHR ICA Programme Pre-doctoral Clinical Academic Fellowships grant number NIHR 300317, as a Masters in Health Research dissertation. Funding for the open access charges for the publication of this protocol was provided by the NIHR Clinical Research Network West Midlands. SS is part funded by the NIHR Applied Research Collaboration (ARC) West Midlands, the NIHR Health Protection Research Unit (HPRU) Gastrointestinal Infections and the NIHR HPRU Genomics and Enabling Data.

**Disclaimer** The views expressed in this publication are those of the author(s) and not necessarily those of the NIHR, HEE, NHS or the UK Department of Health and Social Care.

**Competing interests** None declared.

**Patient and public involvement** Patients and/or the public were involved in the design, or conduct, or reporting, or dissemination plans of this research. Refer to the Methods section for further details.

**Patient consent for publication** Not applicable.

**Ethics approval** Not applicable.

**Provenance and peer review** Not commissioned; externally peer reviewed.

**Data availability statement** All data relevant to the study are included in the article or uploaded as supplementary information. This review presents previously published and publicly available data. Please refer to the reference list (in article) and their authors for these research data.

**ORCID iDs**
Eleanor Jayne Hoverd http://orcid.org/0000-0002-8482-655X
Jeremy Dale http://orcid.org/0000-0001-9256-3553

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
