## [Reviewer comments · BMJ Open]

ARTICLE DETAILS

TITLE (PROVISIONAL)	Factors influencing decisions about whether to participate in health research by people of diverse ethnic and cultural backgrounds: a realist review.
AUTHORS	Hoverd, Eleanor; Hawker-Bond, George; Staniszewska, Sophie; Dale, Jeremy

VERSION 1 – REVIEW

REVIEWER	Shepherd, Victoria Cardiff University, Centre for Trials Research
REVIEW RETURNED	12-Nov-2021

GENERAL COMMENTS	Thank you for the opportunity to review this very interesting paper which explores decision-making about research participation by people from ethnic and diverse backgrounds using a realist review approach. Ensuring that people from ethnic and diverse backgrounds have the opportunity to participate in and benefit from research is a hugely important area to address. The manuscript is well written and makes a useful contribution to the evidence on supporting the inclusion of under-served groups in research. However, some revisions are needed which would strengthen the manuscript, in particular the need to put it into context about decisions to participate in research more broadly and to provide further methodological detail. There are also some suggestions related to text/grammar. Introduction/Background p.5 line 96-97 Suggest it should be either an introduction or a background as currently labelled as both. p.5 line 110 the phrase 'whom are under-served' is used throughout. I am not sure that it is the correct use of 'whom' and suggest 'who are under-served' throughout p.5 lines 115-119 Whilst there is no agreed definition on what under-served means and who under-served groups are, this does not capture the broad range of the types of factors that might contribute to groups being under-served. I suggest revisiting the INCLUDE work cited and rephrasing this to give readers a broader understanding of the topic and that this review focuses on one of these factors, although many will be intersectional. Methods p.6 lines 130-132 Although NIHR is written in full in the article summary, it should be written out in full in the main text. Some
--

	explanation is needed about who the NIHR are (this will not be familiar to an international readership) and who the Research Champions are, what they do etc. A link is provided but a brief sentence would be useful here, and describing the Champions involved in this project. p.7 lines 157-158 Similarly, the term 'stakeholder' is used throughout but there no details about who these were, how they were selected, how many meetings, when the meetings were held etc. Again, although a reference to the protocol is provided, some details need to be added here. In the article summary there is mention of the strength being that meetings were held remotely and so enabled a wider geographical spread but without any information in the text to explain who and how this activity was conducted it isn't supported. P7. Line 160 Literature searches were conducted iteratively – is the text at the end of the previous section about 'see online supplementary 2 searches' meant to be located here? Also, when were the searches conducted? p.8 lines 171-175 refers to the justification for the decision to limit the subsequent searches to the UK as the contexts and resources may differ – however the article summary (although not the main text) states that 'Whilst the review was undertaken in the context of the UK health care system, the programme theory is likely to have wider applicability.' If that is being claimed, then some discussion about this claim and the limits and differences is needed in the Discussion section. P9. Line 208 There is a great deal of detail about the methods used for searching and selection of documents, but no details about the methods used in the synthesis beyond that it was undertaken by one researcher. More detail is needed here about the process and how analytical decisions were made, e.g was there discussion/review/agreement between the research team about CMOCs? P10. Lines 226-230 As previously, more information is needed about the process of developing recommendations – who, how etc. Would also suggest changing the term IRBs (primarily a US term) to RECs as these recommendations are for the UK context. Results P10. Lines 235-239 Given that the source types included a relatively low proportion of primary empirical research, this should be included in the limitations of the review. Were there pragmatic or other reasons why literature and systematic reviews were included – rather than the original papers they reviewed? Was there any overlap e.g the study was included in this review in its own right as well as in one of the reviews and so 'double counted'? Details about this should be added to the Methods section. P12. Lines 290-291 Appears to be a truncated sentence. P14. Lines 331-223 This sentence does not make sense and should be revised for clarity.
--	--

	P.16 Table 1 The first recommendation ‘acting on the issues that we are aware of.’ should be revised for clarity. The third recommendation links tailored information with diverse public involvement – but creating different formats such as video and translation also requires substantial resources. It is not clear if the need for additional resources (e.g in the abstract) includes resources for this type of activity? Reference to this (which is a recommendation to funders) should be considered. Discussion The Discussion should be expanded so that the findings are placed within the context of existing literature around decisions about participating in research. More detail should be included about how the findings from this study add to, or contrast with, existing literature. For example, how do these findings compare with what we know about how broad or less diverse populations decide whether to participate in research? What are the differences (if any)? p.18 lines 387-388 As a lack of diversity amongst researchers is identified as a mechanism (presumably because having diverse perspectives has an impact on the research) it would be useful to reflect on this issue in relation to the research team for this project. P18. Lines 393-394 The term ‘shared decision-making’ is used in this review, and the need for enabling shared decision-making is included in the abstract. In the context of making decisions about research (which is the title and aim of this review) the term is widely meant as meaning the decision to participate in research (or choice about treatment etc) is shared between the patient/participant and the clinician/researcher. See NICE guidelines and patient decision aid literature. If the term is being used differently here (as collaborative decisions about research design etc) then either providing context/a definition for this term or using an alternative term should be considered. Also, in the final programme theory diagram shared decision-making can work positively or negatively – therefore more needs to be said about this if shared decision-making is being encouraged as a result of this review. What might the negative impacts be? How can they be mitigated against? Strengths and limitations See previous comment about UK vs other contexts, and the types of sources included in the review. There is emphasis on the strengths of having more than one reviewer undertaking the screening and assessment of relevance and rigour, but not the potential limitations of only one researcher conducting data synthesis (see previous comment about the need for more detail about this in the Methods section). Supplementary files In the diagram depicting the final programme theory the right-hand blue arrows are quite bold and jumbled up, including obscuring one of the boxes. I suggest revising this to make it a bit clearer.
REVIEWER	Paternotte, Emma Institute for Health Services Utrecht

GENERAL COMMENTS

Review comments

Study: Factors influencing decisions about whether to participate in health research by people of diverse ethnic and cultural backgrounds: a realist review.

Abstract:

Well written, although I miss the reason to perform this research in the introduction.

The conclusion is more a summary. Could you explain in one sentence what the conclusion is of your review.

Introduction/background:

In the introduction of these reviews should the following questions be explained:

What in the background? □ line 98-102

What is the problem? Line 107-109

Why is this important? Line 109-110

What is already known? □ line 102-106

You could think of putting this 4 answers in the above placed sequence. For me it would be easier to read on this way. This method is described by Lorelei Lingard.

Line 111-113: this is described what is done in this review.

However, in my opinion this should not be placed in the introduction/background.

I would advise to add a sentence about the reason to use the realist review method since this not a well known review method for many readers.

Methods:

What is the difference between the scoping search (line 134) and the evidence search (line 159)? If these two are different than why is this?

Supplementary 3: was there no limitation on language of the included studies? Why was there a limitation in the date of 15 years?

Supplementary 4: what do the numbers, CMO1 for example, mean? Is this the first CMO of the 14 extracted CMO's? If yes than please explain this in the text? If no, than please explain what these numbers mean.

Figure 2 is a result. I missed this in the result section.

Results

Here again I missed an overview with the 14? CMO's?

The examples help in understanding the content of the CMO's.

Table 2 is nice to read and very concrete.

Discussion

Nice short summary to start with. However, I miss some more explanation or comparison with other studies or theories about this topic.

Line 390-398 could be placed in the result section since this is something you see in the results. If you think that line 390-398 are not a result of your study than I would expect an interpretation more than a summary of the added references.

	Line 413 is a good explanation of the method of a realist review. It could help the reader if this is also placed in the method section. Lin 417-419 this is not a strength or a weakness
--	--

REVIEWER	Harden, Samantha Virginia Polytechnic Institute and State University, Human Nutrition, Foods, and Exercise
REVIEW RETURNED	01-Dec-2021

GENERAL COMMENTS	Changing the way researchers, IRBs, and other key stakeholders think about and implement structures for engaging folks with diverse ethnic and cultural backgrounds in research is necessary. Recommendations should be based on prior experiences, knowledge, and empirical data, which the authors have conducted and propose within this manuscript. There are some major revisions needed to thread the content and context with the resultant action items, but this work shows promise. Some specific and general feedback is outlined below. There are additional and superfluous spaces throughout the text within the abstract 71 *were* included—then share more about the % across peer review, grey literature, etc. Background needs a wider opening statement... Perhaps and line 99 needs “participation in research” “Our population” who is our? Is this global context or UK specific, if so, state explicitly? The background seems to solely focus on the limitations of consent process rather than the “participant research experience.” Suggest adding more literature related to that (i.e., exploration of CMOCs). For examples, there is ample literature on the “inter-relationships” found in the results, which should be mentioned in the background with existing gaps to inform the theory and actionable steps. Line 128: RAMESES guidance is not universal. Please describe for context Line 131 NIHR not defined in first use within main text Define who the research champions are and what their involvement was Line 134: Remove rough and replace with preliminary or otherwise more precise language Line 147- add information about the CMOCs- their original use, definitions, etc. Line 153- stakeholder involvement might be paired with the Patient Public involvement section above Line 161- what is hand-searching? Should this be manual? Line 183- Authors are using documents and citations (then studies on line 233) interchangeably. Are these different? Should documents be used throughout to represent the diverse types of “reports” used?
--

	Line 185- was inclusion assessed independently then compared or assessed together? And how was the relevance and rigor assessment different from the title and abstract review? Were these sequential? Line 258- define researcher matching Great results and data and the ideas are solid but need refining. Some of the text needs to be tightened up for grammar and prose as well as succinct meaning “acting on the issues that we are aware of.” The actions required should be more explicit as well “raise awareness” what does that mean? What are best practices that are not merely performative? How do you train people within institutions on values and beliefs?
--	---

VERSION 1 – AUTHOR RESPONSE

Reviewer 1	Authors’ response
Dr. Victoria Shepherd, Cardiff University Comments to the Author: Thank you for the opportunity to review this very interesting paper which explores decision-making about research participation by people from ethnic and diverse backgrounds using a realist review approach. Ensuring that people from ethnic and diverse backgrounds have the opportunity to participate in and benefit from research is a hugely important area to address. The manuscript is well written and makes a useful contribution to the evidence on supporting the inclusion of under-served groups in research. However, some revisions are needed which would strengthen the manuscript, in particular the need to put it into context about decisions to participate in research more broadly and to provide further methodological detail. There are also some suggestions related to text/grammar.	Thank you for your encouraging comments, the feedback and suggestions have been addressed as outlined in this point-by-point response.

Introduction/Background	
p.5 line 96-97 Suggest it should be either an introduction or a background as currently labelled as both.	Thank you for pointing this out, “Background” sub-title has been deleted, leaving Introduction as heading (Line 96).
p.5 line 110 the phrase ‘whom are under-served’ is used throughout. I am not sure that it is the correct use of ‘whom’ and suggest ‘who are under-served’ throughout	Thank you, this has been changed in this line and throughout, deleting the term whom and now reading: Line 119 ...by under-served populations Line 118-119 ... “those whom are under-served” deleted, now reads “ under-served groups”
p.5 lines 115-119 Whilst there is no agreed definition on what under-served means and who under-served groups are, this does not capture the broad range of the types of factors that might contribute to groups being under-served. I suggest revisiting the INCLUDE work cited and rephrasing this to give readers a broader understanding of the topic and that this review focuses on one of these factors, although many will be intersectional.	The work published by Witham et al 2020 (INCLUDE project) has been re-visited and thus the following has been rephrased to read (Lines 126-131): There are a number of intersecting factors that contribute to groups being under-served; such as demographic, socioeconomic, health status and disease-specific status, leading to disadvantage and discrimination[1]. However, being under-served is also likely to be context-specific, with key characteristics related to trial design that may affect the ability to participate[1]. For example, the informed consent process has been identified as a specific barrier to inclusion in health research by under-served groups[1].
Methods	
p.6 lines 130-132 Although NIHR is written in full	

in the article summary, it should be written out in full in the main text. Some explanation is needed about who the NIHR are (this will not be familiar to an international readership) and who the Research Champions are, what they do etc. A link is provided but a brief sentence would be useful here, and describing the Champions involved in this project.	NIHR now written out in full (line 157) . Explanation about the NIHR and who Research Champions are, has now been added as follows (lines 157-166): The National Institute for Health Research (NIHR) , established in 2006, funds health and social care research in the UK and advocates for patient and public involvement (PPI) throughout all stages of research. To support PPI, the NIHR have introduced the role of NIHR (Public) Research Champions who are patients or members of the public, and volunteer to raise awareness of health and social care research as well as help researchers to understand the participant experience[14]. Six NIHR (Public) Research Champions were involved in this review, helping to clarify the scope of the review, contributing to the development of programme theory and developing recommendations. Three of the NIHR (Public) Research Champions were from populations of diverse ethnic and cultural backgrounds [14].
p.7 lines 157-158 Similarly, the term ‘stakeholder’ is used throughout but there no details about who these were, how they were selected, how many meetings, when the meetings were held etc. Again, although a reference to the protocol is provided, some details need to be added here. In the article summary there is mention of the strength being that meetings were held remotely and so enabled a wider geographical spread but without any information in the text to explain	Further detail added about stakeholders: Lines 186-189 Stakeholders included NIHR (Public) Research Champions, health professionals (research nurses, a dietician, physiotherapists and a general practitioner) and clinical academics who were openly invited to participate via the review team’s associated Medical School and NIHR Clinical Research Network (CRN).

who and how this activity was conducted it isn't supported.	Details added about meetings, who and how conducted Lines 189-196: A total of six meetings were held virtually (three with NIHR (Public) Research Champions and three with the mixed group of health professionals and clinical academics). Stakeholders were geographically widespread and thus virtual meetings enabled easier access than had meetings been held face to face. At each stakeholder meeting, background information was provided, including an explanation of realist methodology. The facilitated discussions covered under-served populations; initial programme theory; analysis; results and recommendations; and stakeholder feedback and advice.
P7. Line 160 Literature searches were conducted iteratively – is the text at the end of the previous section about 'see online supplementary 2 searches' meant to be located here? Also, when were the searches conducted?	Yes, thank you for highlighting. This has been amended to read : Lines 202-203 and conducted between May through August 2020 (see online supplementary file 2 Searches) In addition, the month/year when searches were conducted has been added to the supplementary file 2 Searches.
p.8 lines 171-175 refers to the justification for the decision to limit the subsequent searches to the UK as the contexts and resources may differ – however the article summary (although not the main text) states that 'Whilst the review was undertaken in the context of the UK health care system, the programme theory is likely to have wider applicability.' If that is being claimed, then	Lines 473-482 , further discussion has been added to support this claim, as follows, in the Discussion section: The programme theory from this realist review is likely to have wider applicability despite being carried out in the context of the UK health care system. The informed consent process is a universal cornerstone to ethical practice in

some discussion about this claim and the limits and differences is needed in the Discussion section.	health research although may vary in the way in which it is carried out. However, rigid study designs and delivery may be contributing to the global issue of under-representation of under-served populations in health research[18,50]. While the critical issues described in this review are likely to be suitable for consideration by health research systems outside of the UK, it is recognised that resources and context differ across research systems, which may limit the transferability, particularly in low-income countries. The recommendations in this realist review evaluation in different settings to understand their wider applicability and transferability.
P9. Line 208 There is a great deal of detail about the methods used for searching and selection of documents, but no details about the methods used in the synthesis beyond that it was undertaken by one researcher. More detail is needed here about the process and how analytical decisions were made, e.g., was there discussion/review/agreement between the research team about CMOCs?	Further detail now provided in the Methods section as follows Lines 145-155: RAMESES guidance has been produced to provide initial reporting standards for realist reviews and evaluations[23]. Following a realist analytical process, Context, Mechanism, Outcome, Configurations (CMOCs) were built by EH. These were shared and discussed at meetings with the review team and stakeholders whereby feedback was encouraged which contributed to further refinement and establishing a long list of CMOCs. The CMOCs conceptualise key features that support explanation and understanding of complex programmes and create a model that describes how mechanisms are activated, for who and under what circumstances, which affects outcomes[23]. Summaries were developed after each meeting to support reflection and understanding with stakeholders. Diagrams were constructed and refined to support explanation of the findings.
P10. Lines 226-230 As previously, more information is needed about the process of	Lines 280-283 now read as follows, with IRBs changed to RECs:

developing recommendations – who, how etc. Would also suggest changing the term IRBs (primarily a US term) to RECs as these recommendations are for the UK context.	Recommendations were developed with stakeholders through presenting the findings and a narrative of the programme theory through a virtual meeting. EH led the remote discussion leading to agreement upon a set of recommendations aimed at government agencies, Research Ethics Committees (RECs), researchers, and health professionals. Input from stakeholders was intended to ensure their practicality.
Results	
P10. Lines 235-239 Given that the source types included a relatively low proportion of primary empirical research, this should be included in the limitations of the review. Were there pragmatic or other reasons why literature and systematic reviews were included – rather than the original papers they reviewed?	This is mentioned in the strengths and limitations (Lines 77-79) : There is a paucity of primary empirical research around the informed consent process in health research with people of diverse ethnic and cultural backgrounds, limiting the evidence from which to extract data for the review. Literature and systematic reviews were included and papers referenced within them, and deemed relevant, according to the Inclusion/Exclusion criteria, were reviewed. These documents helped to develop, refute and refine aspects of programme theory that offered existing data that was able to be turned into mid-range theories suitable for testing against data in other documents outside of those reviews.

Was there any overlap e.g., the study was included in this review in its own right as well as in one of the reviews and so ‘double counted’? Details about this should be added to the Methods section.	Lines 235-241 added in Methods: All relevant data was used to build the arguments that supported the final programme theory, including systematic and literature reviews.[25] Two papers included in the review are noted to be included in a systematic review, but do not impact upon the review in terms of the importance given to CMOC development.[6,41] Documents identified as being helpful in developing, refuting and refining the programme theory were sought, although the number of iterations and length of time spent on this was limited due to the timescale of the review.[25]
P12. Lines 290-291 Appears to be a truncated sentence.	New text added (Lines 345-347) : Contextual factors such as the life situation of some populations (e.g., migrant communities) can influence the decision-making process in health research due to ...fear of identification and reporting to authorities[5](P.13) (CMOC 11).
P14. Lines 331-223 This sentence does not make sense and should be revised for clarity.	Thank you – this now reads (Lines 385-389) : When the needs (e.g., cultural, linguistic or historical) of people from diverse ethnic and cultural backgrounds are misrepresented or ignored [3,5,72] (CMOC 6, CMOC 8, CMOC 13) this results in: discriminatory outcomes (e.g., exclusion due to eligibility criteria) and enforces inappropriate ethical, legislative, scientific guidelines and recruitment processes (CMOC 5, CMOC 13).[3,5,8]
P.16 Table 1	Table 1. (Line 416) First recommendation now reads :

The first recommendation 'acting on the issues that we are aware of.' Should be revised for clarity. The third recommendation links tailored information with diverse public involvement – but creating different formats such as video and translation also requires substantial resources. It is not clear if the need for additional resources (e.g., in the abstract) includes resources for this type of activity? Reference to this (which is a recommendation to funders) should be considered.	Policymakers, organisations, institutions and individuals need to take responsibility and address institutional racism, identifying and tackling issues that contribute to this (e.g., lack of culturally adapted information, practices that may be contributing to institutional racism.) Third recommendation now reads: Develop tailored informed consent in health research that offers choices (e.g., written, audio, multi-media), which may require additional resources.
Discussion	
The Discussion should be expanded so that the findings are placed within the context of existing literature around decisions about participating in research. More detail should be included about how the findings from this study add to, or contrast with, existing literature. For example, how do these findings compare with what we know about how broad or less diverse populations decide whether to participate in research? What are the differences (if any)?	Discussion expanded to read (Lines 420-425): This realist review of 71 sources summarises and expands upon the existing evidence around decision-making and participation in health research through focussing upon the context and mechanisms surrounding decision-making and understanding. The evidence base on participation in health research has tended to focus on the psychosocial drivers of participation, but use of a realist approach enabled greater consideration of contextual factors that influence whether participation occurs.

p.18 lines 387-388 As a lack of diversity amongst researchers is identified as a mechanism (presumably because having diverse perspectives has an impact on the research) it would be useful to reflect on this issue in relation to the research team for this project.	A sentence has been added in response to this reading: Lines 430-434: Whilst the review team itself was not ethnically diverse, the stakeholder group were ethnically, professionally and organisationally diverse. EH is part of an Equality, Diversity, Inclusion working group which supported engagement with patient and public stakeholders from diverse cultural and ethnic backgrounds.
P18. Lines 393-394 The term 'shared decision-making' is used in this review, and the need for enabling shared decision-making is included in the abstract. In the context of making decisions about research (which is the title and aim of this review) the term is widely meant as meaning the decision to participate in research (or choice about treatment etc) is shared between the patient/participant and the clinician/researcher. See NICE guidelines and patient decision aid literature. If the term is being used differently here (as collaborative decisions about research design etc) then either providing context/a definition for this term or using an alternative term should be considered. Also, in the final programme theory diagram shared decision-making can work positively or negatively – therefore more needs to be said about this if shared decision-making is being encouraged as a result of this review. What might the negative impacts be? How can they be mitigated against?	To clarify, this has been explained and more context provided as follows Lines 442-462: The differences and similarities between the use of the term shared decision making in clinical care and within a health research context would benefit from additional exploration of the literature. The term shared decision making is defined by the UK's National Institute for Health and Care Excellence (NICE) shared decision making collaborative as: a process in which clinicians and patients work together to select tests, treatments, management or support packages, based on clinical evidence and the patient's informed preferences. It involves the provision of evidence-based information about options, outcomes and uncertainties, together with decision support counselling and a system for recording and implementing patients' informed preferences.[78-79]

	It is of key importance in protecting patient/participant autonomy and voluntariness.[76-79] In the context of this review, shared decision making is part of the informed consent process and requires establishing understanding by the potential participant, discussion of the risks and benefits, as well as clarifying the consequences of different options available.[77] Shared decision making is also a vital element of co-production, or the involvement of patients and the public in shaping research projects.[80] In some instances, shared decision making can result in negative, or positive outcomes; for example, the hierarchies within some families may control decision-making through preventing or granting permission for family members to take part in health research.[6,9] More positive, shared decision making occurs when researchers spend time with communities, providing opportunities for informal discussions about research, building trust whilst developing cultural sensitivity to the “norms and values” of particular groups.[26]
Strengths and Limitations	
See previous comment about UK vs other contexts, and the types of sources included in the review. There is emphasis on the strengths of having more than one reviewer undertaking the screening and assessment of relevance and rigour, but not the potential limitations of only one researcher conducting data synthesis (see previous comment about the need for more detail about this in the Methods section).	With the additional explanation provided in the stakeholder involvement section, Lines 186-196, this is hoped to have clarified how they were involved in data analysis and synthesis, that potentially improves validity. Further detail has also been added in Lines 147-155.

In the diagram depicting the final programme theory the right-hand blue arrows are quite bold and jumbled up, including obscuring one of the boxes. I suggest revising this to make it a bit clearer.	Figure 4 Final Programme Theory has been amended, reducing the weight line of the arrows, creating a clearer illustration. In addition, the headings within the circles have been enlarged and moved slightly, to make the description within the circles clearer.
--	--

Reviewer 2	
Dr. Emma Paternotte, Institute for Health Services Utrecht	
Abstract Well written, although I miss the reason to perform this research in the introduction. The conclusion is more a summary. Could you explain in one sentence what the conclusion is of your review.	Amended as follows, Lines 49-52: The review indicates the need for a more inclusive research infrastructure that facilitates diverse participation in health research through incorporating adaptive processes that support shared decision-making within the informed consent process and in the conduct of research projects.
Introduction	
In the introduction of these reviews should the following questions be explained: What in the background? ◇ line 98-102 What is the problem? Line 107-109 Why is this important? Line 109-110 What is already known? ◇ line 102-106 You could think of putting these 4 answers in the above placed sequence. For me it would be easier to read on this way. This method is described by Lorelei Lingard.	This has been addressed through re-organising this section and adding in more information about what is known. This section now flows in the order in which you have laid out, as follows: Lines 98-109: People who have poorer health outcomes, and have greater needs are currently under-served by the way in which health research is accessed and delivered[1]. Lack of diversity of participants taking part in health research, poses a serious challenge both within the UK and globally, around how to address and develop a more inclusive health research system so that under-served populations can benefit. It has been argued that the informed consent process may be contributing to inequalities in health research

	through a lack of flexibility in the process[2-10] . The informed consent process in health research aims to provide patients and the public with the information they require to make a voluntary decision about participation in health research. In May 2019, the International Council for Harmonisation (ICH)[11]-drafted the General Considerations for Clinical Studies E8 (R1) suggesting that a “one size fits all approach” to studies should be avoided. Hence, it is important to unravel the factors that may affect the decision to take part in health research by these populations, to develop a system that is more inclusive.
Line 111-113: this is described what is done in this review. However, in my opinion this should not be placed in the introduction/background. I would advise to add a sentence about the reason to use the realist review method since this not a well-known review method for many readers.	Moved this sentence that describes what was done, to Methods : Lines 139-141 In this study, we undertook a realist review to produce a programme theory to explain the informed consent process, and hence provide actionable recommendations for policymakers and health researchers to inform the development of inclusive health research. Added sentence as follows in Introduction: Lines 118--121: Conducting a realist review of the informed consent process in health research with under-served groups allows exploration of this complex procedure[39]. This realist review provides an explanation as to how the informed consent process may work for whom, how and under what circumstances.
Methods	

What is the difference between the scoping search (line 134) and the evidence search (line 159)? If these two are different than why is this?	Clarification around scoping search now provided in Lines 171-172: This allowed us to explore the available literature and to create some initial boundaries[25].
Supplementary 3: was there no limitation on language of the included studies? Why was there a limitation in the date of 15 years?	The search was limited to English documents only as there was no funding for this project for translation. This has been added to Supplementary file 3 as follows: Only sources written in English will be included. In addition , the following justification has been added : The search was limited to a period of 15 years due to the extent to which research infrastructure and research governance has changed over that period of time. Inclusion of review articles (published within the last 15 years) included evidence from earlier studies so that we were able to draw on earlier evidence where it was relevant. In addition to this, the evidence reviewed was sufficient to inform and develop programme theory which was the purpose of the review.
Supplementary 4: what do the numbers, CMO1 for example, mean? Is this the first CMO of the 14 extracted CMO's? If yes than please explain this in the text? If no, then please explain what these numbers mean.	Explanation added under CMOC section Lines 296-299: Patterns in Mechanisms were identified as can be seen on the long list of 29 CMOCs used to develop programme theory (see online supplementary file 4 CMOC Long List). These are discussed below making reference to the associated, numbered CMOCs and selected quotations from documents which help to explain the theories.

Figure 2 is a result. I missed this in the result section.	Figure 2 is related to initial programme theory and not final programme theory, therefore not mentioned in the results section.
Results	
Here again I missed an overview with the CMO's? The examples help in understanding the content of the CMO's. Table 2 is nice to read and very concrete.	Overview added as follows Lines 275-278: An overview of theories is described in the results section, through a narrative of how the informed consent process works, for whom, how and under what circumstances based upon the evidence in this review. References are made to CMOCs , including some quotations to draw out key aspects of CMOCs.
Discussion	
Nice short summary to start with. However, I miss some more explanation or comparison with other studies or theories about this topic. Line 390-398 could be placed in the result section since this is something you see in the results. If you think that line 390-398 are not a result of your study than I would expect an interpretation more than a summary of the added references.	This has been expanded as per Reviewer 1's comments in Lines 420-425 as follows: This realist review of 71 sources summarises and expands upon the existing evidence around decision-making and participation in health research through focussing upon the context and mechanisms surrounding decision-making and understanding. The evidence base on participation in health research has tended to focus on the psychosocial drivers of participation, but use of a realist approach enabled greater consideration of contextual factors that influence whether participation occurs. Through deeper exploration using iterative literature searching, data emerged that indicated the use of complex study documentation e.g., patient information leaflets, communication barriers, lack of social presence in the community by researchers and a lack of diversity amongst researchers as key realist

	mechanisms which led to an explanation of the interaction with context to influence outcomes[2,8,54,72]. The Discussion section follows the sub-heading as per BMJ Open guidance which asks for statement of principal findings.
Line 413 is a good explanation of the method of a realist review. It could help the reader if this is also placed in the method section.	This has been added in the Introduction Lines 119-120.
Lin 417-419 this is not a strength or a weakness	Lines 417-419 deleted (It is hoped that the review will be useful to policymakers, organisations, institutions and health researchers in taking action to make changes to the service, for the benefit of those who need research the most[71].)

Reviewer 3	
Dr. Samantha Harden, Virginia Polytechnic Institute and State University Comments to the Author: Changing the way researchers, IRBs, and other key stakeholders think about and implement structures for engaging folks with diverse ethnic and cultural backgrounds in research is necessary. Recommendations should be based on prior experiences, knowledge, and empirical data, which the authors have conducted and propose within this manuscript. There are some major revisions needed to thread the content and context with the resultant action items, but this work shows promise. Some specific and	Thank you. Superfluous spaces have been removed from the abstract.

general feedback is outlined below. There are additional and superfluous spaces throughout the text within the abstract	
71 *were* included—then share more about the % across peer review, grey literature, etc.	The types of data used in the review are explained in Lines 487, with number and % of types of sources identified : The realist review has drawn from both peer-reviewed literature (66,93%) and grey literature (5, 7%) as well as bringing in expertise from the stakeholder group. Theory in realist research is based upon many arguments and Emmel et al 2018 (Doing Realist Research) advise that a focus on data quality is “too narrow a way of assessing ‘quality’” and individually rating “quality” is not taking a realist approach. “Analysis and interpretations are made from ‘nuggets’ of data that spread across documents, or sources and a pragmatic approach should be taken”. It is also understood that when data is limited, even data that has limited trustworthiness can contribute to building arguments that underpin programme theory.
Background needs a wider opening statement... Perhaps and line 99 needs “participation in research”	Background has been re-organised with further clarification and a wider opening statement provided, as per Reviewer 2’s comments (Lines 97-117). Lines 103-105 revised to read: The informed consent process in health research aims to provide patients and the public with the information they require to make a

	voluntary decision about participation in health research.
“Our population” who is our? Is this global context or UK specific, if so, state explicitly?	Changed to read more clearly (Line 98-101) : Lack of diversity of participants taking part in health research, poses a serious challenge both within the UK and globally, around how to address and develop a more inclusive health research system so that under-served populations can benefit.
The background seems to solely focus on the limitations of consent process rather than the “participant research experience.” Suggest adding more literature related to that (i.e., exploration of CMOCs). For examples, there is ample literature on the “inter-relationships” found in the results, which should be mentioned in the background with existing gaps to inform the theory and actionable steps.	Background has been revised and re-named INTRODUCTION, highlighting an existing gap in the literature (Lines 114-115) .
Line 128: RAMESES guidance is not universal. Please describe for context	This is now described further Lines 143-145 : accordance with RAMESES (Realist and Meta-narrative Evidence Syntheses : Evolving Standards) guidance[13]. This provides reporting standards for realist reviews and evaluations[13].
Line 131 NIHR not defined in first use within main text Define who the research champions are and what their involvement was	These have been addressed as per Reviewer 1’s comments in Lines 159-162 .
Line 134: Remove rough and replace with preliminary or otherwise more precise language	The term “rough” is often used in realist research to describe that initial programme theory, but to ensure consistency and clarity, this has been removed and described as initial programme theory, as follows:

	Line 168 In order to develop an initial programme theory
Line 147- add information about the CMOCs- their original use, definitions, etc.	Further information has been added about CMOCs in Lines 150-155 : The CMOCs conceptualise key features that support explanation and understanding of complex programmes and create a model that describes how mechanisms are activated, for who and under what circumstances, which affects outcomes[23]. Summaries were developed after each meeting to support reflection and understanding with stakeholders. Diagrams were constructed and refined to support explanation of the findings.
Line 153- stakeholder involvement might be paired with the Patient Public involvement section above	The reason for placing Stakeholder Involvement after the initial rough programme theory was to keep the flow of steps in chronological order. Stakeholder involvement followed after initial programme theory was created. Hence, we would prefer not to make this suggested change.
Line 161- what is hand-searching? Should this be manual?	Changed to manual Line 204 : manual-searching
Line 183- Authors are using documents and citations (then studies on line 233) interchangeably. Are these different? Should documents be used throughout to represent the diverse types of “reports” used?	Amended for consistency, to documents only and manuscript checked for any further inconsistencies. Line 287 (previously 183) reads: seventy-one documents
Line 185- was inclusion assessed independently then compared or assessed together? And how was the relevance and rigor assessment different from the title and abstract review? Were these sequential?	Clarified to read as follows, Line 229 : by EH and G H-B, independently

	Further clarification to these 2 questions has been added as follows, Lines 229-230: Relevance and rigour were assessed in parallel. Decisions were based on whether a document was relevant to the over-arching research question and what its contribution would be to the review.
Line 258- define researcher matching	This has been clarified in Lines 317-323: There does not appear to be one definition of researcher-matching; examples include: an interviewer speaking the same language as the participant conducts interviews; a health professional who speaks the same language as participant conducts intervention; an interpreter is present to support delivery of the intervention; trained multi-lingual facilitators; ethnically matched outreach team delivers the intervention; research assistants who speak the same language; and culturally adapted interventions.[2,9,17,37,52-59,53]
Great results and data and the ideas are solid but need refining. Some of the text needs to be tightened up for grammar and prose as well as succinct meaning “acting on the issues that we are aware of.”	Thank you. The manuscript has been improved and the meaning of “acting on the issues that we are aware of” has been expanded in Recommendation 1.
The actions required should be more explicit as well “raise awareness” what does that mean? What are best practices that are not merely performative? How do you train people within institutions on values and beliefs?	“raise awareness” changed to (Recommendation 1 – Action column) : inform and educate staff about institutional racism, what it is and how to address it. Further research on developing an inclusive health research system that will explore the values and beliefs of those who design and deliver health research is a recommendation

	and may help to answer these questions (Lines 529-533).
--	--

VERSION 2 – REVIEW

REVIEWER	Shepherd, Victoria Cardiff University, Centre for Trials Research
REVIEW RETURNED	24-Jan-2022

GENERAL COMMENTS	The authors have considerably revised the manuscript in line with reviewers' comments which has considerably strengthened the paper and I recommend accepting it for publication. A couple of minor grammatical issues (e.g All relevant data was ...') can be addressed during the editorial process.
--